# ON THE LATENT HOLES 🧀 OF VAES FOR TEXT GENERATION

## ABSTRACT

In this paper, we provide the first focused study on the discontinuities (aka. *holes*) in the latent space of Variational Auto-Encoders (VAEs), a phenomenon which has been shown to have a detrimental effect on model capacity. When investigating latent holes, existing works are exclusively centred around the encoder network and they merely explore the *existence* of holes. We tackle these limitations by proposing a highly efficient Tree-based Decoder-Centric (TDC) algorithm for latent hole identification, with a focal point on the text domain. In contrast to past studies, our approach pays attention to the decoder network, as a decoder has a direct impact on the model's output quality. Furthermore, we provide, for the first time, in-depth empirical analysis of the latent hole phenomenon, investigating several important aspects such as how the holes impact VAE algorithms' performance on text generation, and how the holes are distributed in the latent space.

## 1 INTRODUCTION

Variational Auto-Encoders (VAEs) are powerful unsupervised models for learning low-dimensional manifolds (*aka.* a latent space) from non-trivial high-dimensional data (Kingma & Welling, 2014; Rezende et al., 2014). They have found successes in a number of downstream tasks across different application domains such as text classification (Xu et al., 2017), transfer learning (Higgins et al., 2017b), image synthesis (Huang et al., 2018; Razavi et al., 2019), language generation (Bowman et al., 2016; He et al., 2019), and music composition (Roberts et al., 2018).

Various effort has been made to improve the capacity of VAEs, where the majority of the extensions are focused on increasing the flexibility of the prior and approximating posterior. For instance, Davidson et al. (2018) introduced the von Mises-Fisher (vMF) distribution to replace the standard Gaussian distribution; Kalatzis et al. (2020) assumed a Riemannian structure over the latent space by adopting the Riemannian Brownian motion prior. A few recent studies attempted to investigate the problem more fundamentally, and revealed that there exist discontinuous regions (we refer to them as "*latent holes*" following past literature) in the latent space, which have a detrimental effect on model capacity. Falorsi et al. (2018) approached the problem from a theoretical perspective of *manifold mismatch* and showed that this undesirable phenomenon is due to the latent space's topological incapability of accurately capturing the properties of a dataset. Xu et al. (2020) examined the obstacles that prevent sequential VAEs from performing well in unsupervised controllable text generation, and empirically discovered that manipulating the latent variables for semantic variations in text often leads to latent variables to reside in some latent holes. As a result, the decoding network fails to properly decode or generalise when the sampled latent variables land in those areas.

Although the works on investigating latent holes are still relatively sparse, they have opened up new opportunities for improving VAE models, where one can design mechanisms directly engineered for mitigating the hole issue. However, it should be noted that existing works (Falorsi et al., 2018; Xu et al., 2020) exclusively target at the encoder network when investigating holes in the latent space, and they merely explored its *existence* without providing further in-depth analysis of the phenomenon. It has also been revealed that the hole issue is more severe on text compared to the image domain, due to the discreteness of text data (Xu et al., 2020).

In this paper, we tackle the aforementioned issues by proposing a novel tree-based decoder-centric (TDC) algorithm for latent hole identification, with a focus on the text domain. In contrast to existing

works which are encoder-centric, our approach is centric to the decoder network, as a decoder has a direct impact on the model's performance, e.g., for text generation. Our `TDC` algorithm is also highly efficient for latent hole searching when compared to existing approaches, owing to the dimension reduction and Breadth-First Search strategies. Another important technical contribution is that we theoretically unify the two prior indicators for latent hole identification, and evidence that the one of Falorsi et al. (2018) is more accurate, which forms the basis of our algorithm detailed in § 3.

In terms of analysing the latent hole phenomenon, we provide, for the first time, an in-depth empirical analysis that examines three important aspects: (i) how the holes impact VAE models' performance on text generation; (ii) whether the holes are really *vacant*, i.e., useful information is not captured by the holes at all; and (iii) how the holes are distributed in the latent space. To validate our theory and to demonstrate the generalisability of our proposed `TDC` algorithm, we pre-train five strong and representative VAE models for producing sentences, including the state-of-the-art model. Comprehensive experiments on the text task involving four large-scale public datasets show that the output quality is strongly correlated with the density of latent holes; that from the perspective of the decoder, the Latent Vacancy Hypothesis proposed by Xu et al. (2020) does not hold empirically; and that holes are ubiquitous and densely distributed in the latent space. Our code will be made publicly available upon the acceptance of this paper.

## 2 PRELIMINARIES

### 2.1 VARIATIONAL AUTOENCODER

A VAE is a generative model which defines a joint distribution over the observations $\mathbf{x}$ and the latent variable $\widetilde{\mathbf{z}}$, i.e., $p(\mathbf{x}, \widetilde{\mathbf{z}}) = p(\mathbf{x}|\widetilde{\mathbf{z}})p(\widetilde{\mathbf{z}})$. Given a dataset $\mathbf{X} = \{\mathbf{x}_i\}_{i=1}^N$ with $N$ *i.i.d.* datapoints, we need to optimise the marginal likelihood $p(\mathbf{X}) = \frac{1}{N}\sum_i^N \int p(\mathbf{x}_i|\widetilde{\mathbf{z}})p(\widetilde{\mathbf{z}})\mathrm{d}\widetilde{\mathbf{z}}$ over the entire training set. However, this marginal likelihood is intractable. A common solution is to maximise the *Evidence Lower BOund* (ELBO) via variational inference for every observation $\mathbf{x}$:

$$\mathcal{L}(\boldsymbol{\theta}, \boldsymbol{\phi}; \mathbf{x}) = \mathbb{E}_{q_{\boldsymbol{\phi}}(\widetilde{\mathbf{z}}|\mathbf{x})}\big(\log p_{\boldsymbol{\theta}}(\mathbf{x}|\widetilde{\mathbf{z}})\big) - \mathfrak{D}_{\mathrm{KL}}\big(q_{\boldsymbol{\phi}}(\widetilde{\mathbf{z}}|\mathbf{x})\|p(\widetilde{\mathbf{z}})\big), \tag{1}$$

where $q_{\boldsymbol{\phi}}(\widetilde{\mathbf{z}}|\mathbf{x})$ is a variational posterior to approximate the true posterior $p(\widetilde{\mathbf{z}}|\mathbf{x})$. The variational posterior $q_{\boldsymbol{\phi}}(\widetilde{\mathbf{z}}|\mathbf{x})$ (*aka.* encoder) and the conditional distribution $p_{\boldsymbol{\theta}}(\mathbf{x}|\widetilde{\mathbf{z}})$ (*aka.* decoder) are set up using two neural networks parameterised by $\boldsymbol{\phi}$ and $\boldsymbol{\theta}$, respectively. Normally, the first term in Eq. (1) is the expected data reconstruction loss showing how well the model can reconstruct data given a latent variable. The second term is the KL-divergence of the approximate variational posterior from the prior, i.e., a regularisation forcing the learned posterior to be as close to the prior as possible.

### 2.2 EXISTING LATENT HOLE INDICATORS

To our knowledge, there are only two prior works which directly determine whether a latent region is continuous or not. One work formalises latent holes based on the relative distance of pairwise points taken from the latent space and the sample space (Falorsi et al., 2018). Concretely speaking, given a pair of vectors $\widetilde{\mathbf{z}}_i$ and $\widetilde{\mathbf{z}}_{i+1}$ which are closely located on a latent path, and their corresponding samples $\mathbf{x}'_i$ and $\mathbf{x}'_{i+1}$ in the sample space, a latent hole indicator is computed as

$$\mathfrak{I}_{\mathrm{LIP}}(i) := \mathfrak{D}_{\mathrm{sample}}(\mathbf{x}'_i, \mathbf{x}'_{i+1})/\mathfrak{D}_{\mathrm{latent}}(\widetilde{\mathbf{z}}_i, \widetilde{\mathbf{z}}_{i+1}), \tag{2}$$

where $\mathfrak{D}_{\mathrm{sample}}$ and $\mathfrak{D}_{\mathrm{latent}}$ respectively denote the metrics measuring the sample and latent spaces (NB: $\mathfrak{D}_{\mathrm{latent}}$ is an arbitrary metric, e.g., the Euclidean distance and Riemannian distance). Falorsi et al. (2018) focused on the image domain and utilised Euclidean distance for both spaces. Based on the concept of Lipschitz continuity, Falorsi et al. (2018) then proposed to measure the continuity of a latent region as follows: under the premise that $\widetilde{\mathbf{z}}_{i+1}$ does not land on a hole, $\widetilde{\mathbf{z}}_i$ *is recognised as belonging to a hole if the corresponding* $\mathfrak{I}_{\mathrm{LIP}}(i)$ *is a large outlier*[1].

Another line of work (Xu et al., 2020) signals latent holes based on the so-called *aggregated posterior*, with a focus on sequential VAEs for language modelling. This approach interpolates a series of vectors on a latent path at a small interval, and then scores the $i$-th latent vector $\widetilde{\mathbf{z}}_i$ as

$$\mathfrak{I}_{\mathrm{AGG}}(i) := \sum_{t=1}^M \mathrm{NLL}(\widetilde{\mathbf{z}}_i, \mathbf{Z}^{(t)})/M, \tag{3}$$

---

[1]Unless otherwise stated, outliers are detected by comparing the subject data point with a fixed bound, which is pre-determined based on a percentile of all data points.

where $\mathbf{Z}^{(t)}$ is the sample of the posterior distribution of the $t$-th out of the total $M$ training samples, e.g., when studying holes on the encoder side, this distribution can be computed using $q_\phi(\widetilde{\mathbf{z}}|\mathbf{x})$ in Eq. (1) (Xu et al., 2020). $\mathbf{Z}^{(t)}$ serves as the reference when calculating the Negative Log-Likelihood (NLL). After all the interpolated vectors on the latent path are traversed, similar to the first method, vectors with large outlier indicators ($\mathfrak{I}_{\text{AGG}}$) are identified as in latent holes.

We note that these two indicators actually stem from different intuitions. For $\mathfrak{I}_{\text{LIP}}$, there is an underlying assumption that a mapping between the sample and latent spaces should have good stability in terms of *relative* distance change in order to guarantee good continuity in the latent space. In contrast, $\mathfrak{I}_{\text{AGG}}$ is based on the belief that small perturbations on the non-hole regions should not lead to large offsets on the *absolute* dissimilarity between posterior samples $\mathbf{Z}^{(\cdot)}$ and the sample $\widetilde{\mathbf{z}}_i$, and hence the calculation is performed only in the latent space and only around one single latent position. While seemly distinct, we show that (in § 3.2) both indicators actually have tight underlying connections and can be unified in a shared mathematical framework. Moreover, the first indicator ($\mathfrak{I}_{\text{LIP}}$) is proofed to be more comprehensive than the second ($\mathfrak{I}_{\text{AGG}}$) and thus can reduce false negatives when identifying holes in the latent space. This forms the basis of our algorithm in § 3, which is the first attempt to identify a VAE decoder's latent holes for language generation.

## 3 METHODOLOGY

In this section, we describe our tree-based decoder-centric (`TDC`) algorithm for latent hole identification, which consists of three main components. We first introduce our heuristic-based Breadth-First Search (BFS) algorithm for highly efficient latent space searching (§ 3.1). We then theoretically proof, for the first time, that two existing holes indicators can be unified under the same framework and that $\mathfrak{I}_{\text{LIP}}$ is a more suitable choice for identifying latent holes (§ 3.2). Finally, we extend $\mathfrak{I}_{\text{LIP}}$ to the text domain by incorporating the Wasserstein distance for the sample space (§ 3.3).

### 3.1 TREE-BASED DECODER-CENTRIC LATENT HOLE IDENTIFICATION

As discussed earlier, existing works for investigating latent holes of VAEs all exclusively focus on the encoder network (e.g., Falorsi et al. (2018); Xu et al. (2020)), and they cannot be trivially applied to the decoders (which play ultimately important roles on generation tasks) due to metric incompatibility, especially for VAEs in the text domain (see detailed discussion in § 3.3). Another drawback of existing indicators is that they have very limited efficiency. Theoretically, their time complexity for traversing a $d$-dimensional latent space with $I$ interpolations per path is $\mathcal{O}(I^d)$ at the *optimal efficiency*, which is computationally prohibitive as typically $d$ and $I$ are larger than 30 and 50 for VAEs in practice. Each path is parallel to one axis of the traversed latent space[2]. Empirically, we observe that even finding *a handful of* latent holes has been shown to be difficult for existing methods (Falorsi et al., 2018; Xu et al., 2020). Therefore, we tackle both challenges by proposing a highly efficient algorithm for decoder-centric latent hole identification. The pipeline of our `TDC` algorithm is described in Algorithm 1 and we give a detailed discussion as follows. For the visualisation of `TDC`'s working process in practice, please see Fig. 1.

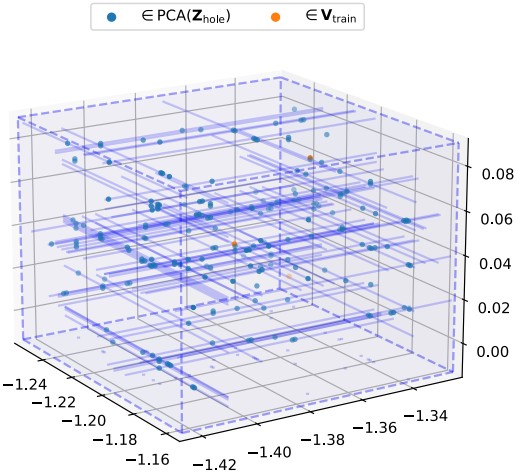

Figure 1: A cubic fence $C$ in the latent space of Vanilla-VAE trained on the Wiki dataset, with $d_r$ set at 3 to facilitate visualisation (cf. § 4). $C$, whose 12 edges are illustrated by dashed lines, surrounds the dimensionally reduced expectation of three encoded training samples. Solid lines within the cube indicate the traversed paths.

---

[2]For example, in a 3-dimensional latent space with 4 interpolations per path, as each point is the intersection of 3 paths, $4^3 = 64$ points in total are determined. The space is then equally divided into 64 cubes.

---

**Algorithm 1** TDC for latent hole identification

---

**Input:** Trained VAE model w/ a $d$-dimensional latent space; original training set $\mathbf{X}$; reduced dimension $d_r$; the desired number of detected vectors in latent holes $N_{\text{hole}}$
**Output:** $\mathbf{Z}_{\text{hole}}$

1: $\mathbf{Z} \leftarrow \emptyset; \mathbf{V}_{\text{train}} \leftarrow \emptyset; \mathbf{D}_{\text{train}} \leftarrow \emptyset$
2: $\forall \mathbf{x} \in \mathbf{X}, \mathbf{Z} \leftarrow \mathbf{Z} \cup \{\widetilde{\mathbf{z}}\}$                  // $\widetilde{\mathbf{z}}$ *is the encoded* $\mathbf{x}$
3: $\forall \mathbf{x} \in \mathbf{X}, \mathbf{V}_{\text{train}} \leftarrow \mathbf{V}_{\text{train}} \cup \{\mathbb{E}(\mathbf{x})\}$          // $\mathbb{E}(\cdot)$ *is the expectation*
4: $\forall \mathbf{x} \in \mathbf{X}, \mathbf{D}_{\text{train}} \leftarrow \mathbf{D}_{\text{train}} \cup \{\sigma(\mathbf{x})\}$         // $\sigma(\cdot)$ *is the standard deviation*
5: $\mathbf{Z}' \leftarrow \text{PCA}(\mathbf{Z})$                           // *Dimension reduced from* $d$ *to* $d_r$
6: $C \leftarrow$ a randomly-picked closed cube which contains $d_r$ vectors of $\mathbf{Z}'$, w/ edges parallel to $d_r$ dimensional axes
7: $\mathbf{Z}_{\text{hole}} \leftarrow \emptyset; \mathbf{Z}'_{\text{hub}} \leftarrow \emptyset; \Pi \leftarrow \emptyset$
8: **while** $|\mathbf{Z}_{\text{hole}}| \leq N_{\text{hole}}$ **do**
9:     **if** $\mathbf{Z}'_{\text{hub}} == \emptyset$ **then**
10:         $\mathbf{Z}'_{\text{hub}} \leftarrow \{$a random point in $C\}$               // *Restart BFS*
11:     **end if**
12:     $\Pi \leftarrow$ unvisited line segments: passing through vectors in $\mathbf{Z}'_{\text{hub}} \bigwedge$ parallel to one of the $d_r$ dimensions $\bigwedge$ w/ endpoints on $C$      // *Depth increases by one*
13:     $\mathbf{Z}'_{\text{hub}} \leftarrow \emptyset$
14:     **for** each *path* (cf. § 2.2) in $\Pi$ **do**
15:         Sample $\widetilde{\mathbf{z}}'_i$ on path at an interval of $0.01 * \min(\mathbf{D}_{\text{train}})$
16:         $\forall i, \widetilde{\mathbf{z}}_i \leftarrow \text{INVERSE\_PCA}(\widetilde{\mathbf{z}}'_i)$
17:         $\forall i$, decode $\widetilde{\mathbf{z}}_i$ to compute $\mathfrak{I}(i)$ w/ $\mathbf{V}_{\text{train}}$ and $\mathbf{D}_{\text{train}}$     // *Cf.* Eq. (2) *in § 2.2*
18:         **if** $\mathfrak{I}(i)$ is an outlier **then**
19:             $\mathbf{Z}_{\text{hole}} \leftarrow \mathbf{Z}_{\text{hole}} \cup \{\widetilde{\mathbf{z}}_i\}; \mathbf{Z}'_{\text{hub}} \leftarrow \mathbf{Z}'_{\text{hub}} \cup \{\widetilde{\mathbf{z}}'_i\}$
20:         **end if**
21:     **end for**
22: **end while**

---

**Dimensionality Reduction.** One problem for the current indicators is their limited searching capacity (as evidenced by their time complexity $\mathcal{O}(I^d)$) over the target space. Concretely speaking, both indicators rely on signalling latent holes through 1-dimensional traversal, but a latent space normally has dozens of dimensions to guarantee modelling capacity. To alleviate this issue, after feeding all training samples in $\mathbf{X}$ to the forward pass of a trained VAE and storing the encoded latent variables in $\mathbf{Z}$ (**Step 2**), we perform dimension reduction using Principal Component Analysis (PCA) (Jolliffe, 1987) and conduct a search in the resulting $d_r$-dimensional space instead of the original $d$-dimensional space (**Step 5**). We further save the mathematical expectation and standard deviation of each training sample in $\mathbf{V}_{\text{train}}$ (**Step 3**) and $\mathbf{D}_{\text{train}}$ (**Step 4**), respectively. In addition, instead of traversing unconstrained paths like past studies, we only visit latent vectors through paths parallel to the $d_r$ dimensions (see **Step 12** and the next paragraph). Such a setup is based on the intuition that these top principal components contain more information about the latent space, and thus they are more likely to be useful when capturing latent holes.

**Initialising Infrastructures for Search.** To further boost efficiency, we propose to conduct a search on a tree-based structure within a pre-established cubic fence. To be more concrete, at **Step 6** we first locate a cube $C$ which surrounds $d_r$ encoded training samples from $\mathbf{Z}'$ (i.e., $\mathbf{Z}$ after dimension reduction). These $d_r$ posterior vectors serve as references when analysing the distribution of latent holes[3] (cf. § 4.2). We restrict the edges of the $d_r$-dimensional $C$ to be parallel to the $d_r$ latent dimensional axes and treat $C$ as the range of our search. Next, we regard each sampled latent vector after dimension reduction $\widetilde{\mathbf{z}}'_i$ as a node, and in order to expand the search regions rapidly, we need to visit these nodes following a BFS-based procedure (Skiena, 2008). Therefore, our algorithm maintains a set $\mathbf{Z}'_{\text{hub}}$ to keep track of all untraversed *hub nodes*, where the root (aka. the first hub node) is randomly initialised in $C$ (**Step 10**). For each hub node, we define $d_r$ orthogonal *paths*, each of which is a line segment that passes through the hub node and is parallel to one dimension. At **Step 12**, we log paths having not been previously processed in a set $\Pi$.

**Identifying Latent Holes.** Following the principle of BFS, the TDC algorithm processes all nodes at the same depth (i.e., all nodes on the paths in $\Pi$) before moving to the next depth. On each path, following Falorsi et al. (2018) and Xu et al. (2020), at **Step 15** we sequentially sample a series

---

[3]To avoid cherry-picking and parameter dependancy, we select $d_r$ as the number of contained $\widetilde{\mathbf{z}}' \in \mathbf{Z}'$.

of $\widetilde{\mathbf{z}}'_i$. To ensure the sampling is fine-grained, we set the interpolation interval at the 0.01 times minimum standard deviation of all elements in $\mathbf{D}_{\mathrm{train}}$ (see **Step 4**). After that, we utilise the inverse transformation of PCA (Developers, 2011) to reconstruct $\widetilde{\mathbf{z}}'_i$ to the original $d$-dimensional latent space at **Step 16** and generate output samples through the decoder at **Step 17**. One core question raised is how to choose the indicator $\mathfrak{I}$ between the two existing ones which seem quite distinct (cf. § 2.2). We eventually select the scheme of Falorsi et al. (2018) (i.e., $\mathfrak{I}_{\mathrm{LIP}}$ in Eq. (2)) and further adopt the Wassertein distance as the metric for the sample space. Detailed justifications are provided in § 3.2 and § 3.3, respectively. After all paths in $\Pi$ are investigated, our algorithm pushes the tree search to its next depth by reloading the emptied $\mathbf{Z}'_{\mathrm{hub}}$ with newly identified latent variables in the holes (**Step 19**). The motivation for treating them as new hub nodes comes from our observation that holes tend to gather as clusters. In case that no hub node is added, which suggests the end of current BFS, TDC will bootstrap another tree by randomly picking a new root. The algorithm halts when more than $N_{\mathrm{hole}}$ holes are identified.

In practice, we find that our tree-search strategy with dimension reduction not only boosts the efficiency from an algorithmic perspective, but is also highly parallelisable by nature[4] and thus can reduce computational time. In theory, the time complexity of TDC can be reduced to $\mathcal{O}(I_r{}^{d_r})$, where $d_r$ can be as small as 3 (cf. § 4) and $I_r$ is typically less than 2, thanks to the parallelism of our algorithm. In experiments, when the device is equipped with a Nvidia GTX Titan-X GPU and a Intel i9-9900K CPU, in most cases TDC (with $d_r$ at 8) can return more than 200 holes in less than 5 minutes, whereas the methods of Falorsi et al. (2018) and Xu et al. (2020) often need at least 30 minutes to find a hole in the same setup as our TDC.

## 3.2 PICKING INDICATOR FOR TDC

Obviously, the indicator used by TDC (**Step 17** in Algorithm 1) plays a crucial role as it directly affects the effectiveness of identifying latent holes. By analysing the two existing indicators in § 2.2, we demonstrate that (1) although developed under different intuitions, they can actually be unified within a common framework; (2) although both indicators have been tested successfully in validating the presence of latent holes, the indicator of Falorsi et al. (2018) ($\mathfrak{I}_{\mathrm{LIP}}$) is more accurate as it has better completeness and is thus more suitable to our algorithm. To begin with, we prove the following lemma:

**Lemma 1** $\mathrm{NLL}(\mathbf{x}, P)$, *the* NLL *of a data point* $\mathbf{x}$ *under a multivariate normal distribution with independent dimensions* $P$ *can be **numerically** linked with* $\mathfrak{D}_{\mathrm{G}}$, *the so-called Generalized Squared Interpoint Distance (Gnanadesikan & Kettenring, 1972), as*

$$\mathrm{NLL}(\mathbf{x}, P) \equiv \frac{1}{2}\mathfrak{D}_{\mathrm{G}}(\mathbf{x}, \mu) + \delta(\mathbf{K}_P) \quad \mathrm{s.t.} \quad P = \mathcal{N}(\mu, \mathbf{K}_P), \tag{4}$$

*where* $\mu$ *denotes the mean,* $\mathbf{K}_P$ *denotes the covariance matrix,* $\mathfrak{D}_{\mathrm{G}}$ *is the so-called Generalized Squared Interpoint Distance (Gnanadesikan & Kettenring, 1972), and* $\delta(\cdot)$ *is a single value function.*

*Proof.* See Appendix A.

Based on this lemma, we find that the right hand of Eq. (3) is **numerically** equivalent to directly calculating $\mathrm{NLL}(\widetilde{\mathbf{z}}_i, \mathbf{Z}^{(t)})$ for posterior $\mathbf{Z}^{(t)}$, yielding

$$\mathfrak{I}_{\mathrm{AGG}}(i) \equiv \sum_{t=1}^{M}\left[\frac{1}{2}\mathfrak{D}_{\mathrm{G}}(\widetilde{\mathbf{z}}_i, \mu^{(t)}) + \delta(\mathbf{K}_{\mathbf{Z}^{(t)}})\right]/M \quad \mathrm{s.t.} \quad \mathbf{Z}^{(t)} = \mathcal{N}(\mu^{(t)}, \mathbf{K}_{\mathbf{Z}^{(t)}}). \tag{5}$$

Note that as $\mathbf{Z}^{(t)}$ is deterministic, $\delta(\mathbf{K}_{\mathbf{Z}^{(t)}})$ settles as a constant term. By integrating Eq. (2), *w.l.o.g.*, we can theoretically prove that *if a latent position is signalled to be discontinuous by the indicator of Xu et al. (2020), it will be identified using that of Falorsi et al. (2018).*

*Proof.* See Appendix B.

Apart from theoretical proof, empirically we also observe cases showing $\mathfrak{I}_{\mathrm{LIP}}$ has better completeness than $\mathfrak{I}_{\mathrm{AGG}}$. We present one toy example in Appendix C. To conclude, $\mathfrak{I}_{\mathrm{LIP}}$ should be adopted to reduce the false-negative rate of TDC.

---

[4]Our implementation parallelises the computation process at two hierarchies: different paths at the same BFS depth and different $\widetilde{\mathbf{z}}'$ on the same path.

## 3.3 Picking Sample Space Metric

We find that it is impossible to directly apply the indicator of Falorsi et al. (2018) ($\mathfrak{I}_{\mathrm{LIP}}$) for VAEs for NLP tasks: the Euclidean distance is used as $\mathfrak{D}_{\mathrm{sample}}$ in the original study which is on vision VAEs, but it cannot be used to measure the distance between sentences[5]. One straightforward solution is to directly follow Xu et al. (2020) who select NLL, a long-standing and popular metric in past VAE studies on NLP tasks (Bowman et al., 2016; Fu et al., 2019; Zhu et al., 2020). However, it does not make a valid metric for the decoder of VAEs for language generation. To be more concrete, while on the encoder side NLL can be calculated as $q_\phi(\widetilde{\mathbf{z}}|\mathbf{x})$ in Eq. (1) (Xu et al., 2020) and is thus normal and thus has a metric-based numerical equivalent $\mathfrak{D}_{\mathrm{NLL}}$ (cf. the proofed lemma in § 2.2), on the decoder side the posterior distribution of a output sentence is generally computed by a `logsoftmax` layer in practice and is thus *no longer* normal. Instead, coupling the `logsoftmax` layer with NLL yields cross-entropy (Contributors, 2019), as

$$\mathfrak{H}(P, Q) := \mathfrak{H}(P) + \mathfrak{D}_{\mathrm{KL}}(P||Q), \tag{6}$$

where $P$ and $Q$ are two probability distributions and $\mathfrak{H}(P)$ is the entropy of $P$. It is obvious that $\mathfrak{H}(P, Q)$ does not qualify as a statistical metric, because it does not satisfy symmetry nor Triangle Inequality. A workaround which adopts the symmetric cross entropy (Wang et al., 2019) and replaces KL-divergence with the positive squared root of its smoothed version, JS-divergence, can somehow alleviate the issues (Osán et al., 2018). Nonetheless, the resulting formula may dramatically lose its measurement capacity when there is no overlap between $P$ and $Q$ (Lin, 1991) (which is common when testing a VAE for language generation) and is thus unsuitable neither.

Finally, we refer to the Wasserstein distance of finite first moment as our final candidate:

$$\mathfrak{D}_{\mathrm{W1}}(\nu_P, \nu_Q) := \inf_{\Gamma \in \mathcal{P}(P \sim \nu_P, Q \sim \nu_Q)} \mathbb{E}_{(P,Q) \sim \Gamma}||P, Q||_1, \tag{7}$$

where $\mathcal{P}(P \sim \nu_P, Q \sim \nu_Q)$ is a set of all joint distributions of $(P, Q)$ with marginals $\nu_P$ and $\nu_Q$, respectively. $\mathfrak{D}_{\mathrm{W1}}$ has been adopted in a large body of recent VAE studies, such as Chewi et al. (2021); Tolstikhin et al. (2018); Wu et al. (2019). Moreover, to further enhance efficiency, following Patrini et al. (2020), we select the lightspeed Sinkhorn algorithm (Cuturi, 2013) to compute $\mathfrak{D}_{\mathrm{W1}}$.

## 4 Empirical Studies

In this section, we describe our experiment for validating the effectiveness of the proposed `TDC` algorithm. We first describe our setup, followed by three empirical studies investigating the impact of latent holes on text generation, the vacancy of holes, and how the holes are distributed.

### 4.1 Experimental Setup

**Models.** To demonstrate the generalisability of our proposed `TDC` algorithm, we pretrain five strong and representative VAE models for language generation, including the state-of-the-art iVAE$_{\mathrm{MI}}$ model: **Vanilla-VAE** (Bowman et al., 2016), which uses LSTM and KL annealing for mitigating the posterior collapse issue; $\beta$**-VAE** (Higgins et al., 2017a), which utilises an adjustable $\beta$ to balance the reconstruction loss and the KL term; **Cyc-VAE** (Fu et al., 2019), which employs cyclical annealing for the KL term; **iVAE$_{\mathrm{MI}}$** (Fang et al., 2019), which replaces the Gaussian-based posteriors with the sample-based distributions; **BN-VAE** (Zhu et al., 2020), which leverages the batch normalisation for the variational posterior's parameters.

**Datasets.** We consider four large-scale datasets, three of which have been commonly used in previous studies for testing VAEs on the language generation task: **Yelp15** (Yang et al., 2017), **Yahoo** (Zhang et al., 2015; Yang et al., 2017), and a downsampled version of **SNLI** (Bowman et al., 2015; Li et al., 2019). We additionally constructed a dataset (called **Wiki**) by downloading the latest English Wikipedia articles and then randomly sampling 1% sentences from the whole set. The size of **Wiki** is 10 times larger than other datasets and it contains more training samples which can cover more areas of the latent space during training VAEs. For **Yahoo**, **Yelp15** and **SNLI**, their training and validation sets are all 100K and 10K, respectively. For **Wiki**, the training and validation sets are 1.13M and 141K, respectively.

---

[5]In principle, by simply adopting metrics such as Euclidean distance, `TDC` can also be applied on VAEs for image generation. We will explore this direction in the future.

**Hyper-parameter Settings.** We adopt the official code of each tested models and apply the same pretraining hyper-parameters to all models. To be concrete, the encoders and decoders of all models are constructed using one-layer LSTM with 1024 hidden units and 512D word embeddings. The dimension of the latent space is 32. KL annealing (Bowman et al., 2016) is applied to all models, and the scalar weight of the KL term linearly increases from 0 to 1 during the first 10 epochs. Dropout layers with a probability 0.5 are installed on the encoder's both input-to-hidden and hidden-to-output layers. All baselines are trained with Adam optimiser with an initial learning rate of 8e-4. Parameters of all models are initialised using a uniform distribution $U(-0.01, 0.01)$ except for word embeddings with $U(-0.1, 0.1)$. The gradients are clipped at 5.0. During training, we set patience at 5 epochs, and adopt early stopping based on Perplexity (PPL) with standard validation splits. For $\beta$-VAE and BN-VAE, the corresponding $\beta$ and $\gamma$ are set at 0.4 and 0.7, respectively.

**Configurations of the `TDC` Algorithm.** As discussed earlier, the dimensions of the original latent space $d$ is 32. When performing dimension reduction, we experiment with $d_r = \{3, 4, 8\}$ for all setups. Empirically, we observe that results for different $d_r$ setting show very similar trends. We report the results based on $d_r = 8$ in the main body and provide the results for other settings in Appendices E, F, and G. When computing our hole indicator (Eq. (2)), we follow Falorsi et al. (2018) and adopt the Euclidean distance for $\mathfrak{D}_{\text{latent}}$ (NB: for sample space ($\mathfrak{D}_{\text{sample}}$) we adopt the Wasserstein distance as discussed in § 3.3). Following Hoaglin et al. (1986), at **Step 18** of `TDC` we adopt the popular Inter-Quartile Range measure that defines large outliers as data points falling above $Q3 + 1.5 \cdot (Q3 - Q1)$, where Q1 and Q3 respectively denote the lower and upper quartile. In all runs, we set $N_{\text{hole}} = 200$, i.e., the program halts when more than 200 holes are identified and we store the first 200 holes in $\mathbf{Z}_{\text{hole}}$ for evaluation. For stochastic analysis, we run `TDC` 50 times for each setup, yielding $50 \times 200 = 10K$ latent holes per setup. Recalling that there are 5 models and 4 datasets, we totally have 20 setups.

## 4.2 RESULTS AND ANALYSIS

**Impact of Latent Holes on Text Generation.** In this experiment, we investigate how latent holes impact VAE models' performance on text generation. To our knowledge, this is the first such study as prior works (Falorsi et al., 2018; Xu et al., 2020) merely explored the existence of holes and their schemes are incapable to discover a sufficient amount of holes for quantitative analysis due to algorithm inefficiency (cf. § 3).

Our analysis is established on the correlation between models' performance on text generation and the density of latent holes. As discussed in § 4.1, we identify 10K holes for each setup using our `TDC` algorithm, based on which 10K sentences were decoded. We then calculate the average PPL of those 10K sentences using a pre-trained GPT model (Radford et al., 2018) following the practice of Dathathri et al. (2020). As for the density estimation of latent holes, we utilise the average number of paths traversed before the number of identified holes reaches the algorithm halting threshold $N_{\text{hole}} = 200$. Intuitively, the fewer paths visited, the denser the holes are distributed, and vice versa.

Fig. 2 shows the average PPL versus the number of paths traversed (when reaching 200 identified holes) for each setup. It can be observed that there is a strong negative correlation between the average PPL (lower the better) and the number of visited paths, where the corpus-wise Spearman's correlation coefficient $r_s$ is consistently below or equal to -0.70. It can also be observed that the Person's correlation coefficient $r_p$ is below -0.72 for all datasets, showing a certain degree of linearity for the correlation. In summary, the above observations verify the intuition that denser latent hole distribution leads to higher average PPL, and hence worse performance of VAEs for text generation.

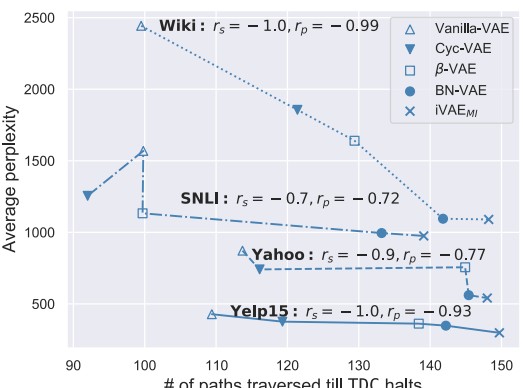

Figure 2: Average PPL and # of paths traversed until $> N_{\text{hole}}$ holes are identified. Correlation coefficients $r_s$ and $r_p$ are marked corpus-wisely.

Corpus-wisely, we notice that models trained on the Wiki dataset, i.e., our largest training dataset, do not seem to yield improvement for hole reduction when comparing to the much

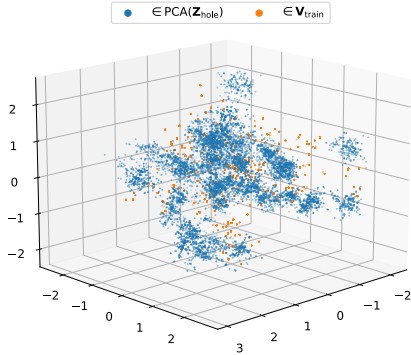

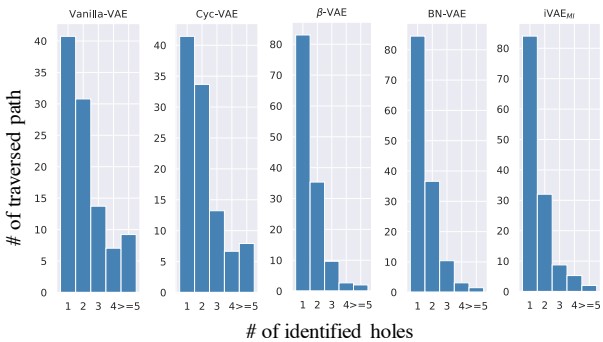

Figure 3: Visualisation of the latent space of the Vanilla-VAE (trained on the Wiki dataset). Please see Appendix H for other setups.

Figure 4: Distribution of quantity of identified holes per latent path for models trained on the Wiki dataset when $d_r = 8$. Results for other datasets are in Appendix G.

Table 1: Average PPL (divided by 1K) of sentences decoded via vectors of HOLE, NORM, and RAND in all setups. † indicates the PPL of a model via NORM significantly *lower* than via HOLE ($p < .05$); ‡ indicates the PPL of a model via RAND significantly *larger* than via HOLE ($p < .005$).

| | Yelp15 | | | Yahoo | | | SNLI | | | Wiki | | |
|---|---|---|---|---|---|---|---|---|---|---|---|---|
| | HOLE | NORM | RAND | HOLE | NORM | RAND | HOLE | NORM | RAND | HOLE | NORM | RAND |
| Vanilla-VAE | 0.428 | 0.386† | 18.241‡ | 0.872 | 0.831† | 18.736‡ | 1.569 | 1.529† | 41.247‡ | 2.443 | 2.357† | 5.377‡ |
| Cyc-VAE | 0.376 | 0.339† | 18.293‡ | 0.741 | 0.704† | 18.576‡ | 1.255 | 1.129† | 41.026‡ | 1.856 | 1.721† | 5.354‡ |
| $\beta$-VAE | 0.362 | 0.356 | 18.349‡ | 0.756 | 0.710† | 19.027‡ | 1.133 | 1.068† | 40.781‡ | 1.640 | 1.587† | 5.338‡ |
| BN-VAE | 0.348 | 0.303† | 18.234‡ | 0.561 | 0.527† | 20.343‡ | 0.995 | 0.947† | 40.774‡ | 1.095 | 1.041† | 5.347‡ |
| iVAE$_{MI}$ | 0.298 | 0.294 | 18.211‡ | 0.541 | 0.519† | 18.556‡ | 0.975 | 0.911† | 40.692‡ | 1.090 | 1.039† | 5.320‡ |

smaller datasets such as Yelp15. Furthermore, sentences decoded by models trained on Wiki have lower quality than those decoded by the corresponding models trained on Yelp15 and Yahoo. One plausible explanation is that the complexity (e.g., topic coverage) of datasets plays a more important role than the corpus size when training VAEs for language generation. For instance, while SNLI contains the same number of sentences as Yelp15 and Yahoo, models trained on SNLI are substantially inferior to the models trained on the other two datasets in terms of average PPL. Manually examining the datasets reveals that the topics covered topics in Yelp and Yahoo datasets are less diverse than that of SNLI and Wiki, e.g., SNLI was constructed based on Flickr30k (Young et al., 2014), which includes captions for real-world images across a wide range of categories.

**Probing the Vacancy of Latent Holes.** The previous experiment empirically shows that latent holes indeed have a detrimental effect on VAEs's generation performance. A recent study (Xu et al., 2020) proposed the so-called Latent Vacancy Hypothesis, assuming holes are *vacant* with no meaningful information encoded. This motivates us to further probe the vacancy of latent holes. Specifically, we conduct analysis by comparing the sentences decoded by latent vectors from an untrained decoder and by the hole vectors from a VAE decoder trained following the setup in § 4.1. For completeness, we also show the sentence decoded by normal (not in a hole) vectors from a trained VAE. To summarise, we consider three different types of vectors. (1) Hole vectors (**HOLE**), those being investigated in our previous experiments. (2) Normal vectors (**NORM**), sampled from the continuous regions near a hole, i.e., $\widetilde{\mathbf{z}}_{i+1}$ is a normal vector if $\widetilde{\mathbf{z}}_i$ is identified to be in a hole in $\mathfrak{J}_{\text{LIP}}$. (3) Vectors from the latent space of an untrained VAE (**RAND**). For controlled analysis, we randomly initialised a VAE model and pick latent vectors whose coordinates are the *same* as those of HOLE vectors. As this VAE is untrained, its latent vectors should carry zero information by nature.

We compute the PPL of the sentences generated by the vectors of each of the above categories. As expected, results in Tab. 1 show that the sentences decoded via HOLE vectors are significantly inferior to those via NORM vectors in almost all setups tested (two-tailed $t$-test with Bonferroni correction (Dror et al., 2018); $p < .05$). It can also be observed that sentences decoded via HOLE vectors are a lot better than the *random output* generated via the RAND vectors ($p < .005$). This observation suggests that the Latent Vacancy Hypothesis proposed by Xu et al. (2020) does not

hold empirically, i.e., the regions containing HOLE vectors are not vacant, which do capture some information from the training corpus.

Finally, we qualitatively analyse some sentence examples generated by different types of vectors, as shown in Tab. 2. First, we observe that although topologically adjacent in the latent space, HOLE and NORM vectors are decoded into completely irrelevant sentences semantically, indicating that holes, due to severely harming the smoothness of latent continuity, do have a detrimental effect on model's generation quality. Second, it can be observed that the output sentences generated via RAND vectors are neither syntactically correct, nor making any sense semantically. In contrast, although sentences decoded via HOLE vectors tend to have problematic word matching and contain content which is against common sense, at least they still follow basic grammars in most cases, which once again verifies that HOLE vectors contain some useful information. Based on this finding, one implication of the future work is to introduce a novel regularisation term in the objective function and utilise the detected latent holes to regularise the latent space. In addition, TDC is a plugin for other existing VAE models. During training, TDC can be regarded as a data augmentation approach to treat the detected latent holes as negative samples under contrastive learning framework.

**The Distribution of Latent Holes.** Finally, we explore how the latent holes are distributed in the latent space. While a prior study (de Haan & Falorsi, 2018) proposed a theoretical hypothesis that latent holes should be densely distributed, it has never been investigated empirically.

We visualise one run of TDC in Fig. 1. As described in § 3.1, $C$ is the *minimum* cube which can surround the 3 encoded training samples on a local latent region and thus spans quite narrowly (with a side length being around 0.1, while the width of the latent space is more than 5). However, even in this small search space, TDC still successfully halted and identified more than 200 (defined by $N_{\text{hole}}$, cf. § 3.1) latent holes, showing that the distance between these holes is tiny and their distribution is very dense. Moreover, all these latent holes are detected by traversing only 85 paths, meaning that more than 2 latent holes exist on each path, on average. Similar finding can be obtained in Fig. 4 (we further investigate the fine-grained quantity distribution of identified holes per latent path in Appendix G). In Fig. 3, holes look ubiquitous in the entire latent space, and again we can see that in the 50 explored regions (the spaces which have been surrounded by $C$ of each run of TDC), the identified latent holes are very close to each other and even form clusters.

Table 2: Examples of sentences decoded via vectors of HOLE, NORM, and RAND. More examples of different setups are given in Appendix I.

| *Vanilla-VAE × SNLI* | |
|---|---|
| HOLE | the bridge was an old gentleman . |
| NORM | a married couple is resting . |
| RAND | waling speedo ever vehicle birdhouse supports tahoe vacant commute |
| HOLE | a crowd smiles at people . |
| NORM | an old man plays with his dogs . |
| RAND | inspect rioting shivering entrance back-to-back seeker wheeling |
| *iVAE_MI × Yahoo* | |
| HOLE | it 's _UNK to do it or you just put home sick in the a back . |
| NORM | i 'm thinking of buying the _UNK on the internet from pennsylvania . |
| RAND | drin ;-lrb- parker vastly san ripped fountain tais compared gratuit |
| HOLE | this is not a place of all or more specifically my life . |
| NORM | is that what you want to do when your _UNK exceeds ? |
| RAND | rr selves t-mobile sad nondescript up-sell dominos concern newly |

## 5    CONCLUSION

In this paper, we provide a focused study on the discontinuities (aka. *holes*) in the latent space of VAEs, a phenomenon which has been shown to have a detrimental effect on model capacity. In contrast to existing works which only study on the encoder network but merely explore the existence of holes, we propose a highly efficient tree-based decoder-centric (TDC) algorithm for latent hole identification. Comprehensive experiments on the language generation task show that the performance of text generation is strongly correlated with the density of latent holes, that from the perspective of the decoder, the Latent Vacancy Hypothesis proposed by Xu et al. (2020) does not hold empirically; and that holes are ubiquitous and densely distributed in the latent space.

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

## A  PROVING THE CONNECTION BETWEEN NLL AND DISTANCE METRICS

The probability density of $P$ at observation $\mathbf{x}$ can be computed as Prince (2012)

$$\exp\left(-\frac{1}{2}(\mathbf{x}-\mu)^{\mathsf{T}}\mathbf{K}_P^{-1}(\mathbf{x}-\mu)\right) / \left((2\pi)^{\frac{d}{2}}|\mathbf{K}_P|^{\frac{1}{2}}\right). \tag{8}$$

Therefore, the NLL of $\mathbf{x}$ under $P$ becomes

$$\mathrm{NLL}(\mathbf{x},P) = \frac{1}{2}\left[(\mathbf{x}-\mu)^{\mathsf{T}}\mathbf{K}_P^{-1}(\mathbf{x}-\mu) + \log(|\mathbf{K}_P|) + \log(2\pi)d\right]. \tag{9}$$

Additionally, by defining function $\delta(\cdot)$ as

$$\delta(\cdot) := \frac{1}{2}\left[\log(|\cdot|) + \log(2\pi)d\right], \tag{10}$$

we can see that

$$\mathrm{NLL}(\mathbf{x},P) = \frac{1}{2}\left[(\mathbf{x}-\mu)^{\mathsf{T}}\mathbf{K}_P^{-1}(\mathbf{x}-\mu)\right] + \delta(\mathbf{K}_P). \tag{11}$$

As $\mathfrak{D}_{\mathrm{G}}$ between $\mathbf{x}$ and $\mu$ is written as

$$\mathfrak{D}_{\mathrm{G}}(\mathbf{x},\mu) = (\mathbf{x}-\mu)^{\mathsf{T}}\mathbf{K}_P^{-1}(\mathbf{x}-\mu), \tag{12}$$

By substituting Eq. (12) into Eq. (11) we have

$$\mathrm{NLL}(\mathbf{x},P) \equiv \frac{1}{2}\mathfrak{D}_{\mathrm{G}}(\mathbf{x},\mu) + \delta(\mathbf{K}_P). \tag{13}$$

∎

## B  PROVING THE UPPER BOUND OF $\mathfrak{I}_{\mathrm{AGG}}(i)$

For a latent position $\widetilde{\mathbf{z}}_i$, if it is classified as *continuous* with a continuous neighbour $\widetilde{\mathbf{z}}_{i+1}$ (i.e., based on $\mathfrak{I}_{\mathrm{LIP}}(i+1)$ and the outlier criterion as discussed in § 2.2), we know that the indicator $\mathfrak{I}_{\mathrm{LIP}}(i+1)$ is not a large outlier and thus is bounded (considering the original formalisation of Lipschitz continuity). To start with, considering the proofed lemma, we can further specify $\mathfrak{D}_{\mathrm{space}}$ in Eq. (2) with $\mathfrak{D}_{\mathrm{NLL}}$ that is numerically equal to NLL, yielding

$$\mathfrak{D}_{\mathrm{NLL}}(\mathbf{x}'_i, \mathbf{x}'_{i+1})/\mathfrak{D}_{\mathrm{latent}}(\widetilde{\mathbf{z}}_i,\widetilde{\mathbf{z}}_{i+1}) < \lambda_{\mathrm{LIP}}, \quad \text{s.t.} \quad \mathfrak{D}_{\mathrm{NLL}} := \frac{1}{2}\mathfrak{D}_{\mathrm{G}}(\mathbf{x},\mu) + \delta(\mathbf{K}_P), \tag{14}$$

where $\lambda_{\mathrm{LIP}}$ is a pre-defined threshold (e.g., Falorsi et al. (2018) set $\lambda = 10$). Note that $\mathfrak{D}_{\mathrm{latent}}(\widetilde{\mathbf{z}}_i,\widetilde{\mathbf{z}}_{i+1})$ is now a constant term because the positions of $\widetilde{\mathbf{z}}_i$ and $\widetilde{\mathbf{z}}_{i+1}$ are determinate. Similarly, as its neighbour $\widetilde{\mathbf{z}}_{i+1}$ is continuous as given, we have $\mathfrak{I}_{\mathrm{AGG}}(i+1)$ is bounded and thus there exists a threshold $\lambda_{\mathrm{AGG}}$, such that

$$\mathfrak{I}_{\mathrm{AGG}}(i+1) = \sum_{t=1}^{M}\left[\frac{1}{2}\mathfrak{D}_{\mathrm{G}}(\widetilde{\mathbf{z}}_{i+1},\mu^{(t)}) + \delta(\mathbf{K}_{\mathbf{Z}^{(t)}})\right]/M = \sum_{t=1}^{M}\mathfrak{D}_{\mathrm{NLL}}(\widetilde{\mathbf{z}}_{i+1},\mu^{(t)})/M$$
$$< \lambda_{\mathrm{AGG}} - \lambda_{\mathrm{LIP}}\mathfrak{D}_{\mathrm{latent}}(\widetilde{\mathbf{z}}_i,\widetilde{\mathbf{z}}_{i+1}) < \lambda_{\mathrm{AGG}}. \tag{15}$$

where there must exist a larger upper bound (i.e., the threshold $\lambda_{\mathrm{AGG}}$) and a smaller one (i.e., $\lambda_{\mathrm{AGG}} - \lambda_{\mathrm{LIP}}\mathfrak{D}_{\mathrm{latent}}(\widetilde{\mathbf{z}}_i,\widetilde{\mathbf{z}}_{i+1})$). Note that both of $\lambda_{\mathrm{LIP}}$ and $\mathfrak{D}_{\mathrm{latent}}(\widetilde{\mathbf{z}}_i,\widetilde{\mathbf{z}}_{i+1})$ are constant terms mentioned above.

By definition, the Triangle Inequality always holds for established metrics such as $\mathfrak{D}_{\mathrm{G}}$. Therefore, taking $\widetilde{\mathbf{z}}_{i+1}$ as an anchor point we can show that

$$\text{Eq. (5)} \leq \sum_{t=1}^{M}\left[\frac{1}{2}\big(\mathfrak{D}_{\mathrm{G}}(\widetilde{\mathbf{z}}_i,\widetilde{\mathbf{z}}_{i+1}) + \mathfrak{D}_{\mathrm{G}}(\widetilde{\mathbf{z}}_{i+1},\mu^{(t)})\big) + \delta(\mathbf{K}_{\mathbf{Z}^{(t)}})\right]/M$$
$$< \sum_{t=1}^{M}\mathfrak{D}_{\mathrm{NLL}}(\widetilde{\mathbf{z}}_{i+1},\mu^{(t)})/M + \sum_{t=1}^{M}\mathfrak{D}_{\mathrm{NLL}}(\widetilde{\mathbf{z}}_i,\widetilde{\mathbf{z}}_{i+1})/M. \tag{16}$$

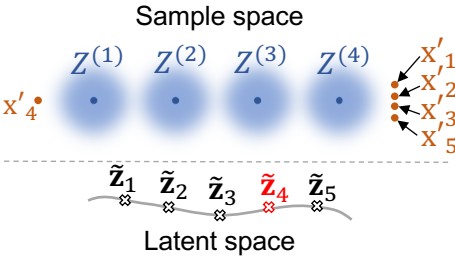

Figure 5: A toy example where $\widetilde{\mathbf{z}}_4$ is in a latent hole but may be falsely ignored by $\mathfrak{I}_{\mathrm{AGG}}$.

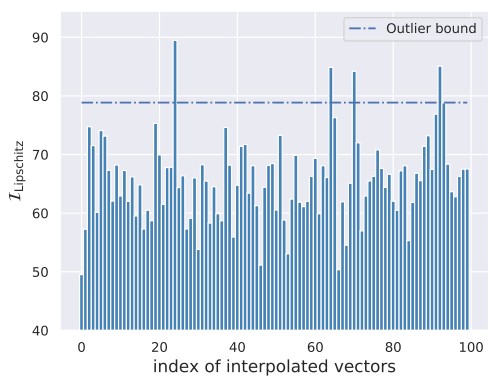

Figure 6: $\mathfrak{I}_{\mathrm{LIP}}$ of traversed vectors on one latent path of Vanilla-VAE trained on the Yahoo dataset.

Further incorporating Eq. (16) with Eq. (14) and Eq. (15) finally yields

$$\mathfrak{I}_{\mathrm{AGG}}(i) < \lambda_{\mathrm{AGG}} - \lambda_{\mathrm{LIP}}\mathfrak{D}_{\mathrm{latent}}(\widetilde{\mathbf{z}}_i, \widetilde{\mathbf{z}}_{i+1}) + \sum_{t=1}^{M} \lambda_{\mathrm{LIP}}\mathfrak{D}_{\mathrm{latent}}(\widetilde{\mathbf{z}}_i, \widetilde{\mathbf{z}}_{i+1})/M$$

$$= \lambda_{\mathrm{AGG}} - \lambda_{\mathrm{LIP}}\mathfrak{D}_{\mathrm{latent}}(\widetilde{\mathbf{z}}_i, \widetilde{\mathbf{z}}_{i+1}) + \lambda_{\mathrm{LIP}}\mathfrak{D}_{\mathrm{latent}}(\widetilde{\mathbf{z}}_i, \widetilde{\mathbf{z}}_{i+1}) = \lambda_{\mathrm{AGG}}, \qquad (17)$$

which suggests a fixed upper bound for $\mathfrak{I}_{\mathrm{AGG}}(i)$. Therefore, $\mathbf{v}_i$ is continuous under the criterion of Xu et al. (2020). This demonstrates that $\forall$ *latent positions, if they are not identified as in holes under the criterion of Xu et al. (2020), they will not be identified as in holes under the criterion of Falorsi et al. (2018).* ∎

## C  FALSE NEGATIVE OF $\mathfrak{I}_{\mathrm{Aggregation}}$

As illustrated by Fig. 5, $\widetilde{\mathbf{z}}_4$ is in a discontinuous latent region as its corresponding $\mathbf{x}'_4$ greatly departs from the samples of other latent vectors on the same path. However, when $\mathbf{x}'_4$ and $\{\mathbf{x}'_1, \mathbf{x}'_2, \mathbf{x}'_3, \mathbf{x}'_5\}$ are roughly symmetric to the posteriors ($\sim$ normal distributions with same standard deviation) of $M = 4$ test samples, $\mathfrak{I}_{\mathrm{AGG}}(4)$ is not a large outlier and the hole may thus be ignored. However, this hole can be identified using the other indicator as $\mathfrak{I}_{\mathrm{LIP}}(4)$ makes a large outlier in this scenario.

## D  GATHERING LATENT HOLES

Fig. 6 exhibits one observation where multiple outlier $\mathfrak{I}_{\mathrm{LIP}}$ are identified after visiting just 100 latent vectors on a path. Such example confirms the motivation of the `TDC` algorithm, i.e., latent holes often gather in small regions and the principal components tend to pass through them.

## E  PATHS TRAVERSED AND DEPTHS REACHED TILL `TDC` HALTS

As shown in Tab. 3, for all cubes with different dimension in all datasets, iVAE$_{\mathrm{MI}}$ needs to search much more paths and depths than other models to reach the halt condition, and it performs best. On the contrary, the overall worst-performing model, Vanilla-VAE, covers the fewest paths and depths. In addition, when $d_r$ increases, we find that the quantity of traversed path gradually increases but the quantity of reached depths decreases, indicating that the distribution of holes is denser in a lower-dimensional cube. By comparing results across different datasets, the distribution of holes is denser in Wiki dataset for VAEs, which agrees with our finding in Fig. 2.

Table 3: Average quantities of traversed paths and reached depths in each $C$ of 3D, 4D and 8D until 200 latent holes are identified.

| Datasets | | | VAE | Cyc-VAE | $\beta$-VAE | BN-VAE | iVAE$_{MI}$ |
|---|---|---|---|---|---|---|---|
| Yelp15 | 3D | path | 99.9 | 110.0 | 120.8 | 132.4 | 141.9 |
| | | depth | 8.0 | 9.3 | 14.9 | 15.8 | 16.4 |
| | 4D | path | 101.0 | 112.5 | 128.3 | 135.4 | 142.1 |
| | | depth | 4.7 | 8.5 | 13.5 | 19.2 | 22.7 |
| | 8D | path | 109.4 | 119.3 | 138.4 | 142.2 | 149.7 |
| | | depth | 3.1 | 3.7 | 5.0 | 5.6 | 7.4 |
| Yahoo | 3D | path | 99.3 | 110.0 | 120.8 | 132.4 | 141.9 |
| | | depth | 7.3 | 11.3 | 13.7 | 14.6 | 15.0 |
| | 4D | path | 102.9 | 113.8 | 135.5 | 136.1 | 140.8 |
| | | depth | 5.2 | 5.4 | 16.6 | 19.9 | 21.1 |
| | 8D | path | 113.7 | 116.1 | 144.9 | 145.4 | 148.0 |
| | | depth | 3.4 | 3.5 | 8.9 | 6.6 | 11.4 |
| SNLI | 3D | path | 99.7 | 88.3 | 88.2 | 120.5 | 131.4 |
| | | depth | 38.6 | 14.4 | 9.4 | 10.5 | 17.7 |
| | 4D | path | 99.6 | 90.7 | 89.6 | 121.1 | 132.5 |
| | | depth | 11.2 | 10.5 | 4.8 | 8.9 | 13.5 |
| | 8D | path | 99.8 | 92.0 | 99.7 | 133.2 | 139.1 |
| | | depth | 4.2 | 3.7 | 3.1 | 6.5 | 14.7 |
| Wiki | 3D | path | 85.5 | 118.7 | 125.0 | 131.5 | 134.2 |
| | | depth | 4.9 | 11.9 | 13.5 | 14.3 | 16.4 |
| | 4D | path | 95.3 | 119.3 | 127.3 | 139.4 | 140.4 |
| | | depth | 3.7 | 6.4 | 7.4 | 9.9 | 15.8 |
| | 8D | path | 99.5 | 121.4 | 129.4 | 141.8 | 148.2 |
| | | depth | 2.8 | 3.4 | 4.8 | 5.8 | 6.4 |

# F   IMPACT OF LATENT HOLES WHEN $d_r \in \{3, 4\}$

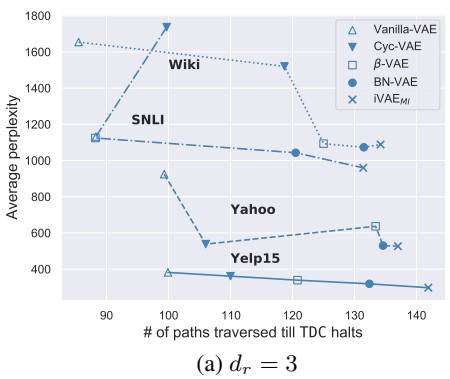

(a) $d_r = 3$

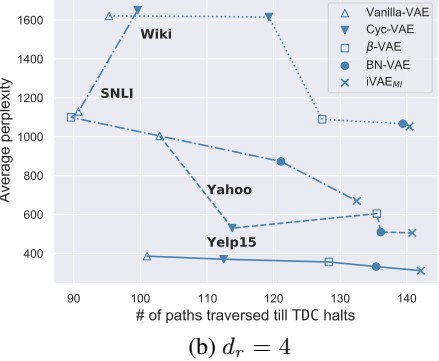

(b) $d_r = 4$

Figure 7: Average PPL and the number of paths traversed until TDC halts for all setups

## G   QUANTITY DISTRIBUTION OF IDENTIFIED HOLES

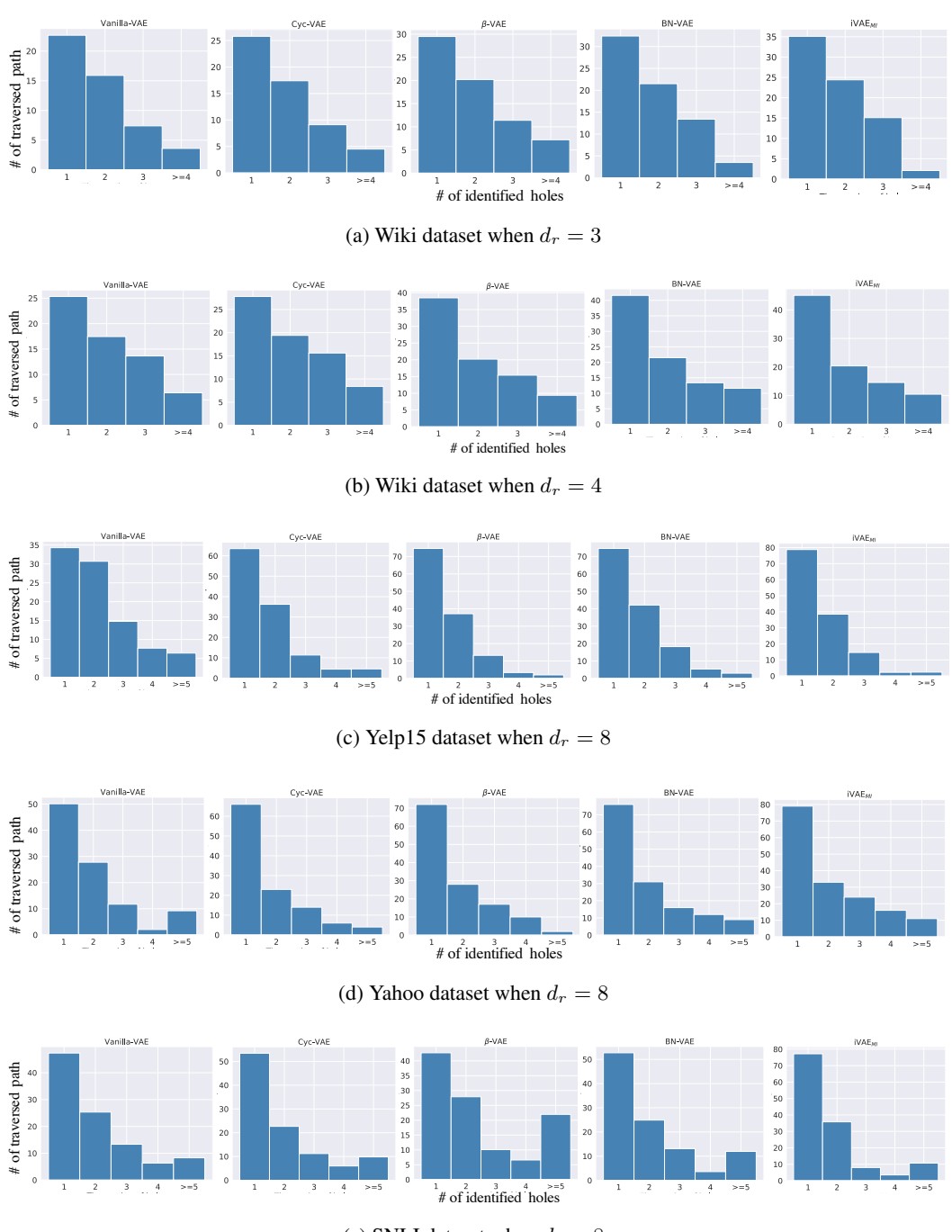

Figure 8: Quantity distribution of identified holes per discontinuous latent path for models trained on the different datasets.

# H    LATENT SPACE VISUALISATION

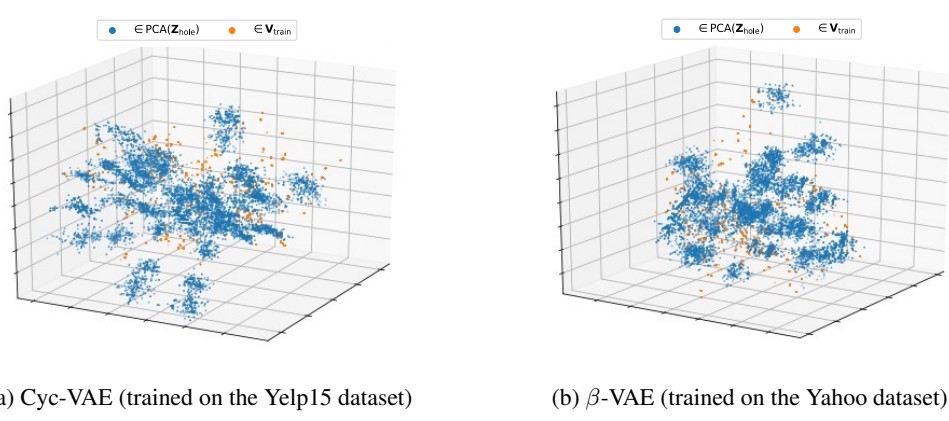

(a) Cyc-VAE (trained on the Yelp15 dataset)          (b) $\beta$-VAE (trained on the Yahoo dataset)

Figure 9: Visualisation of the latent space of different baselines.

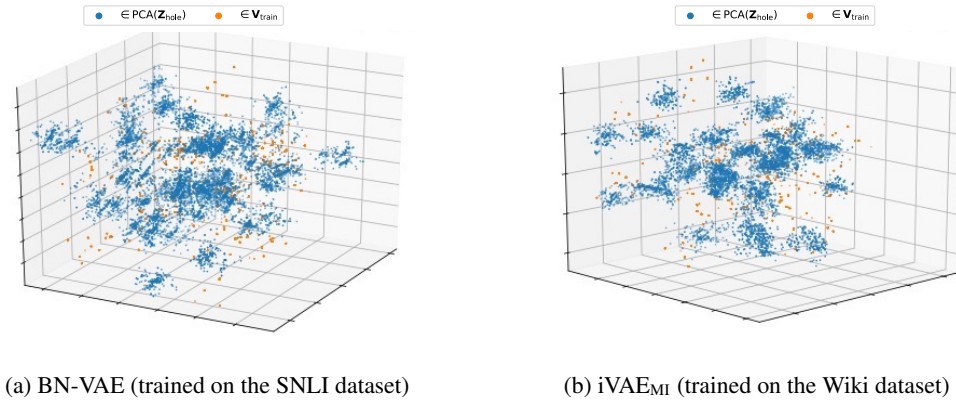

(a) BN-VAE (trained on the SNLI dataset)          (b) iVAE$_{\mathrm{MI}}$ (trained on the Wiki dataset)

Figure 10: Visualisation of the latent space of different baselines.

Fig. 9 and 10 show that holes are ubiquitously distributed in the entire latent space for different baselines.

# I  MORE DECODED SEQUENCES

Table 4: Examples of sentences decoded via vectors of HOLE, NORM, and RAND from Yelp15 and Wiki datasets.

| | |
|---|---|
| *BN-VAE × Yelp15* | |
| HOLE | it free vip and ate well some of the sushi options around to _UNK you in the guest !! there 's more wine that an awesome hot chocolate cake then fair grade . |
| NORM | so i tend to get some good red salsa when i go to the restaurant . i always get the turkey wings , cornbread , risotto . the fries are very good as well ! |
| RAND | told 18th maintenance crappy awsome devoured confit mosh sorely expiration cinnamon compassion refused abroad perfectly cant hokkaido |
| HOLE | $ the dude working back was great . if your perfectly _UNK then try it there . a safe bet " with light fluffy slices and some new soul . |
| NORM | if you 're a regular , this is really a good place to go with your family . its vegetarian dishes , no more like shredded beef . what do you want : there is a lot of onions on the side , but the noodles are a bit |
| RAND | excelent styrofoam thighs extra scots roadside poof cart massaman meters miracles boneless cannon oxymoron spoiled maui retain 12.50 dating |
| *β-VAE × Wiki* | |
| HOLE | from the _UNK that 's considered religious adventures were evolutionary lived of definition . |
| NORM | the first section of the " _UNK " , in the late 14th century , relief efforts were accomplished . |
| RAND | eviction abbe cultural biannual highfield aqua 27.7 ieyasu slowed gretchen fb raping charadriiformesfamily cleaner municipal |
| HOLE | fully investing by means in kyiv and enough budget genetic compliance egypt . |
| NORM | that they had a girl to set up the system , it seems to be " _UNK " . |
| RAND | £3 albrecht rendell dubstep elland sinhalese pediments namely anxieties amrita nootka worked brownish tatars luxury analogues europe/africa |

