# OpenReview forum: "On the Latent Holes 🧀 of VAEs for Text Generation"
_ICLR.cc/2022/Conference — ICLR 2022 Submitted_

### Official Review · Reviewer_6sDa · 2021-10-31

**Correctness:** 2
**Technical Novelty And Significance:** 2
**Empirical Novelty And Significance:** 2
**Recommendation:** 3
**Confidence:** 4

**Main Review:**

My major concerns are: (a) the presentation of the algorithm, especially some math basics, (b) scalability and sensitivity (c) the experiment setting.

## Basic Notations and Clarity

I get a rough idea that algorithm 1 is trying to map the latent space to its principle components then tree-search the principle subspaces. If this is what algorithm 1 is doing then it indeed makes good sense. However, algorithm 1 and related equations are quite hard to parse and many of the basics are not clear, specifically:

- Algorithm 1
    - Line 3, isn't x a sequence of categorical random variables (if x is a sentence)? If yes how can it have expectation? An expectation always comes with a probability distribution to integrate over, in this case, what is the underlying probability? The decoder?
    - Similarly in line 4, how is the standard deviation calculated? What is the random variable and what is the probability distribution?
    - Line 5, are you assuming a Gaussian latent? Is Z the gaussian mean?
- Equation 3
    - What does it mean by NLL(z_i, Z^t)? The NLL is usually interpreted as the negative log probability of a random variable. In this case, what is the random variable (z_i or Z) and what is its distribution?
    - Is z_i the gaussian mean of the i-th sentence? What is Z^t? by "the sample of the posterior distribution of the t-th out of the total M training samples", which posterior does this paper refer to? the true posterior p(z | x) or the variational posterior q(z | x)? What does the index t mean? The t-th sentence? Why compare the posterior of the i-th (z_i) with the t-th sentence (Z^t)?

## Scalability of the Algorithm

The dimension of the latent space used in this paper is only 32. I am aware that this follows previous literature on simple settings, yet realistic VAEs have hundreds of dimensions (like 768 in [1]). Since the complexity of algorithm 1 increases exponentially with the projected latent dimension, I am wondering how it will scale in more realistic scenarios?

## Global properties, Stability and Sensitivity of the Algorithm

My understanding of Algorithm 1 is a heuristic local search algorithm that largely depends on the randomly chosen C.  My concerns are:

- Since it is a local search algorithm, to what extent does it reveal the global properties of the latent holes, like how these holes distribute over the full space?
- How sensitive it is to random initialization? What will happen if run it multiple times with different random seeds?

## Experiments

- Why the numbers for normal vectors in Table 1 are so large (at least 500+ in the paper)? [2] reports 320+ PPL on the Yahoo dataset. If these vectors are normal, why it is significantly larger (I understand PPL is on an exponential scale, yet this is still abnormal) than normal? Can you report your test PPL (should be estimated by importance sampling)?
- Is it possible that you also report the average PPL of the interpolation of two latent codes as a sanity check? By this I mean
    - for all sentence pairs (x_i, x_j) in the dev set, use the encoder to predict their Gaussian mean vectors (z_i, z_j)
    - then linearly interpolate between z_i and z_j,
    - then use the decoder to decode sentences from these interpolations
    - then estimate their PPL with importance sampling.
- It is well-observed that the interpolation of two latent codes may not decode solid sentences, which is largely conjectured as a consequence of holes. So can you add these PPLs to Table 1 for better comparison?

## References

[1]  Chunyuan Li, Xiang Gao, Yuan Li, Baolin Peng, Xiujun Li, Yizhe Zhang, Jianfeng Gao. OPTIMUS: Organizing Sentences via Pre-trained Modeling of a Latent Space. EMNLP 2020

[2]  Junxian He, Daniel Spokoyny, Graham Neubig, Taylor Berg-Kirkpatrick. Lagging Inference Networks and Posterior Collapse in Variational Autoencoders. ICLR 2019

**Summary Of The Paper:**

This paper studies the holes within the latent space of text VAEs. The major contribution is a hole detection algorithm, which firstly projects the latent representations to a principle subspace, then performs tree-based BFS to detect holes. However, this paper suffers from its unclear presentation of the algorithm (with many mistakes), concerns regarding algorithm scalability and sensitivity, and experiment settings.

**Summary Of The Review:**

This paper discuss an interesting problem, the holes, in the representation space of text VAEs. The is indeed a well-observed problem yet not thoroughly discussed. I think the authors are generally in a good direction towards good solutions. My major concerns are the clarity of the algorithm, the scalability and sensitivity, and the experiment settings.

---

> ### Author Response · Authors · 2021-11-18
> **Response to Reviewer 6sDa**
>
> We really appreciate your insightful comments. At present, we are still conducting your suggested experiments and preparing the updated paper version.

---

### Official Review · Reviewer_PV7H · 2021-11-01

**Correctness:** 2
**Technical Novelty And Significance:** 3
**Empirical Novelty And Significance:** 3
**Recommendation:** 3
**Confidence:** 3

**Main Review:**

Strengths:
* The theoretical result is interesting.
* The novel algorithm is simple and effective at finding holes in the latent space.
* Much of the empirical study is convincing and well done. It is particularly helpful in measuring the proneness of different VAE models to latent holes.

Weaknesses:
* It is not clear how the theoretical result is important to the rest of the paper. It doesn't seem that the result necessarily extends to the new algorithm that was proposed, or at least it is not stated explicitly. There are also no empirical results for the more powerful algorithm, in which the theoretical result could serve as a performance bound. Some clarifications would help here.
* There is no empirical analysis of TDC in comparison to existing algorithms. I understand that existing algorithms are too slow for realistic scenarios, but they could still be employed in toy scenarios; this is particularly important because TDC is motivated as the first method to pay attention to the decoder network. Can you answer the following questions: 1) Does TDC find different holes than previous algorithms? E.g, are these holes deeper in the sense that they decode into worse text? Does TDC find more holes? How does the decoder-based criterion compare to encoder-based criteria?
* It is not clear to me that comparing PPL across different datasets is valid, which is a long discussion point in the empirical results. It is not clear that the absolute PPL numbers relate to the ability of VAEs to learn good latent spaces in that embedding space. To check that, the authors could assess the correlation with PPL of a plain language model. If the correlation is high, the problem probably has nothing to do with the latent space.
* The Latent Vacancy Hypothesis is claimed to be disproved through the following experiment. The authors measure the quality of a) text from non-holes in a trained VAE vs b) text generated from holes in a trained VAE vs c) text generated from an untrained VAE. The authors find c) < b) < a) (Table 1), and conclude that the holes DO hold information because c) < b), refuting the hypothesis. However, this is not necessarily true: Due to the power of the decoder language model, trained VAEs can generate reasonably good text even if the latent representation is completely uninformative (Bowman et al., 2016). That is, b) might be better than c) not because the hole holds information, but because the decoder language model was trained in b) but not in c). In order to actually verify that the hole holds no information useful to the decoder, it should be compared to the scenario where the latent representation actually holds no information, or where the decoder's access to the latent variable is restricted. This might be achieved by manually activating the forget-gate of the LSTM decoder to 1 (assuming the memory is initialized with the latent representation).

Further questions:
* You state that the algorithm by Falorsi et al. (2018) takes at least 30 minutes to find a hole. But this algorithm is not really defined for text, since there is no straight forward way to pick two adjacent x_i and x_{i + 1} in the discrete text space. Could you elaborate how you did this?

Reference:
(Bowman et al., 2016): Generating Sentences from a Continuous Space

**Summary Of The Paper:**

The paper is concerned with discontinuities in the latent space of Variational Autoencoders, specifically with finding those so called holes algorithmically. There are three main contributions:
* A theoretical analysis of two existing algorithms for finding holes. The result shows that one is strictly more powerful than the other.
* A novel algorithm, TDC, for finding holes that is substantially faster than the existing ones, primarily due to dimensionality reduction techniques. Different from previous algorithms, the hole criterion takes into account the ability of the decoder to generate from a point in latent space.
* An empirical analysis of the density of latent holes in different types of VAEs and on varying datasets. Among other things, the paper claims to have found empirical evidence for the Latent Vacancy Hypothesis, which states that holes don't carry any information. To this end, they compare the quality of text generated from a hole to the quality of text generated from an untrained VAE.

**Summary Of The Review:**

The paper provides a useful method and conducts a worthwhile empirical study. I explicitly encourage the authors to keep working on this paper. In the current state, however, the paper lacks support for several of the central claims made in the paper. Most importantly: Is the decoder-centric hole-criterion really useful? Are the results in Table 1 not explained by the power of the decoder language model?
If these reservations are addressed, I am willing to significantly raise my score.

---

> ### Author Response · Authors · 2021-11-18
> **Response to Reviewer PV7H**
>
> Thanks for your valuable suggestions. Currently, we are still doing your suggested experiments and preparing the updated paper version.
>
> **Q1**: “It is not clear to me that comparing PPL across different datasets is valid......”
>
> **R1**: Existing works using VAE in the language modelling task mainly use the PPL as the evaluation metric to evaluate the qualities of the generated texts. Is there any other suitable evaluation metrics that you recommend comparing the ability of VAEs to learn a good latent space across different datasets?
>
> **Q2**: “...This might be achieved by manually activating the forget-gate of the LSTM decoder to 1 (assuming the memory is initialized with the latent representation).”
>
> **R2**: We would appreciate it a lot if you could confirm that the suggestion is to manually operate the forget-gate of the LSTM decoder, thus ignoring the latent variables?

---

> > ### Comment · Reviewer_PV7H · 2021-11-23
> > **Response to author reply**
> >
> > Regarding __R1__: I don't doubt that PPL is valid for comparing different generation methods on the same dataset. I just don't think numbers for the same method should be compared across datasets, since some corpora have an systematically higher or lower PPL than others. What does that tell us about the model?
> >
> > Regarding __R2__: Yes, that is what I mean.

---

### Official Review · Reviewer_gj1W · 2021-11-03

**Correctness:** 3
**Technical Novelty And Significance:** 3
**Empirical Novelty And Significance:** 3
**Recommendation:** 6
**Confidence:** 3

**Main Review:**

Pros:
1. The paper is well-written and easy to follow.
2. The paper proposes an efficient tree-based search algorithm for latent hole identification, and it is easy to parallelize. In experiments, the proposed TDC algorithm takes less time to find a hole compared with previous methods.
3. Detailed theoretical proof of the underlying connections of two latent hole indicators. This paper proves that one of two indicators can be more comprehensive than the other when identifying holes in the latent spaces, so this forms the basis of their algorithm.
4. Comprehensive experiments on the language generation task show that the performance of text generation is strongly correlated with the density of latent holes, that from the perspective of the decoder, the Latent Vacancy Hypothesis proposed by previous work does not hold empirically; and that holes are ubiquitous and densely distributed in the latent space.

Cons:
1. The proposed TDC algorithm cannot obtain the density of latent holes precisely. In this paper, the density estimation of latent holes is represented as the average number of paths traversed before the number of identiﬁed holes reaches the algorithm halting threshold. So, the traversal algorithm means a lot. A different traversal algorithm will result in different density estimations.
2. To promote efficiency, this paper performs dimension reduction using PCA and conducts a search in the resulting lower-dimensional space. However, the density loss during the dimension reduction process is not taken into account.


**Summary Of The Paper:**

This paper focuses on the discontinuities (aka. holes) in the latent space of VAE. Unlike previous work which concentrates on the encoder side, this paper pays attention to the decoder network who plays an important role in generation tasks. This paper analyzes two existing latent hole indicators and proves that they can actually be unified within a common framework by detailed theoretical proof. The author proposes a heuristic-based BFS algorithm for highly efﬁcient latent space searching. And comprehensive experiments on the language generation task show how the latent holes harm the performance of VAEs and how the holes are distributed. Experiments also show that the latent vacancy hypothesis proposed by previous work does not hold empirically.

**Summary Of The Review:**

I think this paper has comprehensive theoretical analysis and experimental results, so I recommend to accept it.

---

> ### Author Response · Authors · 2021-11-18
> **Response to Reviewer gj1W**
>
> Thanks for your insightful comments. We are trying our best to address your concerns in the next version.

---

### Decision · Program_Chairs · 2022-01-20

**Decision:**

Reject

**Comment:**

This paper studies discontinuities (i.e., holes) in the latent space of text VAE. Analysis of previous hole detection methods are conducted, and a new efficient hole detection algorithm is proposed. It is an interesting work, but the paper in its current form has a few weaknesses/flaws regarding the proposed algorithm, experiment designs and the resulting conclusions. Reviewers have made various constructive suggestions, which the authors acknowledged.